

# Can simple models predict large scale surface ocean isoprene concentrations?

Dennis Booge[1], Christa A. Marandino[1], Cathleen Schlundt[1], Paul I. Palmer[2], Michael Schlundt[1], Elliot L. Atlas[3], Astrid Bracher[4,5], Eric S. Saltzman[6], Douglas W. R. Wallace[7]

[1]GEOMAR Helmholtz Centre for Ocean Research Kiel, Germany
[2]School of GeoSciences, The University of Edinburgh, Edinburgh, Great Britain
[3]Rosenstiel School of Marine and Atmospheric Science (RSMAS), University of Miami, Florida, USA
[4]Alfred Wegener Institute - Helmholtz Centre for Polar and Marine Research, Bremerhaven, Germany
[5]Institute of Environmental Physics, University Bremen, Germany
[6]Department of Earth System Science, University of California, Irvine, California, USA
[7]Department of Oceanography, Dalhousie University, Halifax, Canada

*Correspondence to*: Dennis Booge (dbooge@geomar.de)

**Abstract**

We use isoprene and related field measurements from three different ocean data sets together with remotely sensed satellite data to model global marine isoprene emissions. We show that using monthly mean satellite derived chl-*a* concentrations to parameterize isoprene with a constant chl-*a* normalized isoprene production rate underpredicts the measured oceanic isoprene concentration by a mean factor of 19±12. Improving the model by using phytoplankton functional type dependent production values and by decreasing the bacterial degradation rate of isoprene in the water column results in only a slight underestimation (factor 1.7±1.2). We calculate global isoprene emissions of 0.21 Tg C for 2014 using this improved model, which is twice the value calculated using the original model. Nonetheless, the sea-to-air fluxes have to be at least one order of magnitude higher to account for measured atmospheric isoprene mixing ratios. These findings suggest that there is at least one missing oceanic source of isoprene influencing the atmospheric concentrations and, therefore, effecting the importance of marine derived isoprene as a precursor to remote marine boundary layer particle formation.

## 1 Introduction

Remote marine boundary layer aerosol and cloud formation are important for both the global climate system/radiative budget and for atmospheric chemistry (Twomey, 1974) and have been investigated, with contentious results, for decades. The question remains: what are the precursors to aerosol and cloud formation over the ocean? Earlier studies pinpointed dimethylsulfide (DMS) as the main precursor, as described in the CLAW hypothesis (Charlson et al., 1987). More recently, this hypothesis has been debated controversially (Quinn and Bates, 2011), because primary organic aerosols (POA; O'Dowd et al., 2008) and small sea salt particles (Andreae and Rosenfeld, 2008;de Leeuw et al., 2011) have been identified as CCN precursors with higher CCN production potential than DMS. In addition to POA, other gases besides DMS have been hypothesized as important for remote marine secondary organic aerosol formation (SOA), including isoprene (2-methyl-1,3-butadiene), which has received the most attention in recent years (Carlton et al., 2009).

Isoprene is a byproduct of plant metabolism and one of the most abundant of the atmospheric volatile non-methane hydrocarbons (NMHC). On a global basis, as much as 90% of atmospheric isoprene comes from terrestrial plant emissions (400-600 Tg C yr-1; Guenther et al., 2006;Arneth et al., 2008). Isoprene is very short-





lived in the atmosphere, with a lifetime ranging from minutes to a few hours. The principal loss mechanism is reaction with hydroxyl radicals (OH), but reactions with ozone and nitrate radicals are also important sinks (Atkinson and Arey, 2003;Lelieveld et al., 2008).

The importance of the ocean as a source of atmospheric isoprene is unclear, as only few studies have directly
measured isoprene concentrations in the euphotic zone. Throughout most of the world oceans, near surface seawater isoprene concentrations range between <1-200 pmol L$^{-1}$, depending on season and region (Bonsang et al., 1992;Milne et al., 1995;Broadgate et al., 1997;Baker et al., 2000;Matsunaga et al., 2002;Broadgate et al., 2004;Zindler et al., 2014;Ooki et al., 2015). Higher isoprene levels have been measured in Southern Ocean and Arctic waters (395 and 541 pmol L-1, respectively; Kameyama et al., 2014;Tran et al., 2013).  Atmospheric
isoprene levels can be as high as 300 parts per trillion (ppt), varying with location and time of day (Shaw et al., 2010). Generally, the mixing ratios are lower than 100 ppt in remote areas not influenced by terrestrial sources (Yokouchi et al., 1999), but can also increase up to 375 ppt during a phytoplankton bloom (Yassaa et al., 2008). Matsunaga et al. (2002) found that the sea-to-air flux estimated from measurements could not explain the atmospheric concentrations observed in the western North Pacific. This agrees with the model calculations of Hu
et al. (2013), who found that top-down and bottom up models estimating isoprene emissions disagree by two orders of magnitude.

Assessing the importance of isoprene for marine atmospheric chemistry and SOA formation requires extrapolations of measurements to develop global emissions climatologies and inventories. Model studies suggest that oceanic sources of isoprene are too weak to control marine SOA formation (Spracklen et al.,
2008;Arnold et al., 2009;Gantt et al., 2009;Anttila et al., 2010;Myriokefalitakis et al., 2010) and field studies indicate that the organic carbon (OC) contribution from oceanic isoprene is less than 2% and out of phase with the peak of OC in the Southern Indian Ocean (Arnold et al., 2009). In contrast, Hu et al. (2013) found that, despite sometimes low isoprene fluxes calculated by models, oceanic isoprene emissions can increase abruptly in association with phytoplankton blooms, resulting in regionally and seasonally important isoprene-derived SOA
formation. Further experiments showed that isoprene oxidation products can increase the level of CCN when the number of CCN is low (Ekstrom et al., 2009). Lana et al. (2012) used both model calculated fluxes of isoprene and remote sensing products to investigate isoprene derived SOA formation in the marine atmosphere. Their results illustrated that the oxidation products of marine trace gases seemed to influence the condensation growth and the hygroscopic activation of small primary particles. Fluxes of isoprene (and other marine derived trace
gases) showed greater positive correlations with CCN number and greater negative correlations with aerosol effective radius than POA and sea salt over most of the world's oceans.

Since isoprene concentration measurements from the open ocean are sparse, it is essential to combine laboratory and field measurements, remote sensing, and modeling if we want to understand marine isoprene emissions. This study utilizes measurements of surface ocean isoprene and associated biological and physical parameters on
three oceanographic cruises to refine and validate the model of Palmer and Shaw (2005) for estimating marine isoprene concentrations and emissions.  The resulting model, with satellite derived input, is used to compute monthly climatologies and annual average estimates of isoprene in the world ocean.

## 2 Methods

### 2.1 Model description





In this study we use a simple steady-state model for surface ocean isoprene consisting of a mass balance between biological production, chemical and biological losses, and emission to the atmosphere (Palmer and Shaw, 2005),

$$P - C_W \left( \sum k_{CHEM,i} C_{Xi} + k_{BIOL} + \frac{k_{AS}}{MLD} \right) - L_{MIX} = 0, \qquad (1)$$

where biological production ($P$) is balanced by all loss processes. $C_W$ is the seawater concentration of isoprene, $k_{CHEM}$ is the chemical rate constant for all possible loss pathways ($i$) with all reactants ($X$) ($X$=OH and $O_2$), $k_{BIOL}$ is the biological loss rate constant, which takes into account the biodegradation of isoprene, $k_{AS}$ is the air-sea gas transfer coefficient that considers the loss processes due to air-sea gas exchange scaled with the depth of the ocean mixed layer (MLD), and $L_{MIX}$ is the loss due to physical mixing (Table 1). The model equation was rearranged to solve for $C_W$ (2) as follows:

$$C_W = \frac{P - L_{MIX}}{\sum k_{CHEM,i} C_{Xi} + k_{BIOL} + \frac{k_{AS}}{MLD}} \qquad (2)$$

The air-sea flux of isoprene ($F$) was calculated using the equation:

$$F = k_{AS}(C_W - C_A / K_H) = \sim k_{AS}(C_W), \qquad (3)$$

where $C_A$ is the air-side concentration of isoprene, and $K_H$ is the dimensionless form of the Henry's law constant (equilibrium ratio of $C_A$ and $C_W$). $C_A$ is assumed to be negligible compared to $C_W$ as noted above (3). As a result, the air-sea isoprene gradient is assumed equal to the surface ocean isoprene level, and emissions are assumed to be first order in $C_W$. This assumption is justified over the open ocean because of the short atmospheric lifetime of isoprene. In coastal regions downwind of strong isoprene sources, this assumption may not be valid. The air-sea exchange transfer coefficient ($k_{AS}$) is computed using the Wanninkhof (1992) wind speed ($U_{10}$) based parameterization and the Schmidt number $S_C$ of isoprene (Palmer and Shaw, 2005):

$$k_{AS} = 0.31 \, U_{10}^2 \left( \frac{S_C}{660} \right)^{-0.5} \qquad (4)$$

Further details about the rate constants and input parameters are described in Table 1. Monthly mean wind speed ($U_{10}$) and sea surface temperature (SST) were obtained from the Quick Scatterometer (QuickSCAT) satellite and the moderate resolution imaging spectroradiometer (MODIS) instrument onboard the Aqua satellite, respectively, and from *in situ* shipboard measurements. MLDs were obtained from climatological monthly means (de Boyer Montégut et al., 2004) and compared to those calculated by *in situ* conductivity, temperature, and depth (CTD) profile measurements during each cruise. MLD was defined as the depth at which temperature is at least 0.2 C higher or lower than the temperature at 10 m depth (de Boyer Montégut et al., 2004). Chlorophyll *a* (chl-*a*) concentrations were obtained either from the MODIS instrument onboard the Terra satellite or from *in situ* shipboard measurements (here chl-*a* is defined as the sum of monovinyl-chl-*a*, divinyl-chl-*a* and chlorophyllide-*a*). Model calculations were carried out using MATLAB (Mathworks).

The steady state model assumption is justified by the relatively short lifetime of isoprene in seawater as air-sea exchange is the dominant loss term over all latitudes and seasons (lifetime: 7-14 days) followed by $k_{BIOL}$ and $k_{CHEM}$ (Palmer and Shaw, 2005). In this study, model runs were carried out using three different sets of model parameters (Table 1):

1) ISO$_{PS05}$ - This is the original configuration used by Palmer and Shaw (2005). In this configuration, the production of isoprene is parameterized as the product of the bulk chl-*a* concentration and a chl-*a*





normalized isoprene production rate ($P_{chloro}$) inferred from laboratory phytoplankton monocultures of several cyanobacteria, eukaryotes, and coccolithophores (Shaw et al., 2003). This approach inherently assumes that all phytoplankton have the same isoprene production characteristics. Palmer and Shaw (2005) also assumed that biological degradation of isoprene occurs in the water column, based on indirect evidence of a biological sink for isoprene (Moore and Wang, 2006), but no isoprene loss rate constants have been published to date. They assumed a global average lifetime of ~17 days ($k_{BIOL}=0.06$ day$^{-1}$) based on the biological degradation rates of different data sets of methyl bromide (Tokarczyk et al., 2003;Yvon-Lewis et al., 2002).

2) ISO$_{PFT}$ - In this configuration different $P_{chloro}$-values are applied for different phytoplankton functional types (PFTs). Laboratory studies have shown that isoprene production rates vary significantly across different PFTs (Bonsang et al., 2010;Colomb et al., 2008;Exton et al., 2013;Shaw et al., 2003;Arnold et al., 2009). We use the PFT-dependent isoprene production rate constants and field observations of PFT distributions to estimate isoprene production rates. The chl-$a$ normalized isoprene production rates of the different algae species are averaged within each PFT to obtain an estimated $P_{chloro}$-value of isoprene for each PFT. PFT distributions along our cruise tracks were derived from the soluble organic pigment concentrations obtained from filtered water samples through Whatman GF/F filters using high pressure liquid chromatography (HPLC) according to the method of Barlow et al. (1997). This method was adjusted to our temperature-controlled instruments as detailed in Taylor et al. (2011). We determined the list of pigments shown in Table 2 of Taylor et al. (2011) and applied the method of Aiken et al. (2009) for quality control of the pigment data. Pigment data from expedition ANT-XXV/1 have been already published in Taylor et al. (2011) and are available from PANGAEA (https://doi.pangaea.de/10.1594/PANGAEA.819070). From the HPLC pigment concentration we calculated PFT groups using the diagnostic pigment (DP) analysis developed by Vidussi et al. (2001) and adapted in Uitz et al. (2006) to relate the weighted sum of seven, for each PFT representative DP. By that the chl-$a$ concentrations for diatoms, dinoflagellates, haptophytes, chrysophytes, cryptophytes, cyanobacteria (excluding prochlorophytes), and chlorophytes were derived. The chl-$a$ concentration of prochlorophytes was derived directly from the divinyl-chl-$a$ concentration (the marker pigment for this group).

3) ISO$_{PFT-kBIO}$ - This configuration utilizes the PFT approach to parameterize isoprene production as in ISO$_{PFT}$ and assumes that biological losses of isoprene in the water column are significantly slower than assumed by Palmer and Shaw (2005). Seawater incubation experiments carried out in temperature controlled water baths over periods ranging from 48-72 hours under natural light conditions, using deuterated isoprene (isoprene-d5), showed significantly longer lifetimes (manuscript in preparation). In the ISO$_{PFT-kBIO}$ configuration, we test a biological degradation lifetime of minimum 100 days ($k_{BIOL}=0.01$ day$^{-1}$).

**2.2 Cruise tracks**

Isoprene was measured in the surface seawater during three separate cruises, the ANT-XXV/1 in the eastern Atlantic Ocean, the SPACES/OASIS cruises in the Indian Ocean and the ASTRA-OMZ cruise in the eastern Pacific Ocean. ANT-XXV/1 took place in November 2008 on board the R/V Polarstern from Bremerhaven, Germany to Cape Town, South Africa (Figure 1, for details about isoprene and ancillary data see also Zindler et al. (2014)). The SPACES/OASIS cruises took place in June/July 2014 on board the R/V Sonne from Durban, South Africa via Port Louis, Mauritius to Malé, Maldives and the ASTRA-OMZ cruise took place in October



2015 on board the R/V Sonne from Guayaquil, Ecuador to Antofagasta, Chile (Figure 1). Air mass backward trajectories (12 hours; starting altitude: 50 m) from the Hybrid Single-Particle Lagrangian Integrated Trajectory (HYSPLIT; http://www.arl.noaa.gov/HYSPLIT.php) model were calculated for the sampling sites. The trajectories, in combination with atmospheric measurements, suggest that the air masses encountered on these

cruises were from over the ocean for more than 12 h prior to sampling, and are therefore unlikely to contain significant isoprene derived from terrestrial sources (Figure 1).

### 2.3 Isoprene measurements

#### 2.3.1 East Atlantic Ocean

The isoprene measurements from the ANT-XXV/1 (November 2008, East Atlantic Ocean) cruise are described

in detail in Zindler et al. (2014). Seawater from approximately 2 m depth was continuously pumped on board and flowed through a porous Teflon membrane equilibrator. Isoprene was equilibrated by using a counter-flow of dry air and was measured using an atmospheric pressure chemical ionization mass spectrometer (mini-CIMS), which consists of a $^{63}$Ni atmospheric pressure ionization source coupled to a single quadrupole mass analyzer (Stanford Research Systems, SRS RGA200). Isoprene from a standard tank was added to the equilibrated air

stream every 12 h to calibrate the system. The precision for isoprene measurements was ± 13%. The isoprene data used here are 5 min averages.

#### 2.3.2 Indian and east Pacific Oceans

The isoprene measurements on the SPACES/OASIS (June/July 2014, Indian Ocean) and ASTRA-OMZ (October 2015, East Pacific Ocean) cruises have not been published previously. Water samples (50 mL) were

taken every three hours from a continuously running seawater pump system located in the ship's moon pool at approximately 6 m depth. All samples were analysed on board within 15 minutes of collection using a purge and trap system attached to a gas chromatograph/mass spectrometer operating in single ion mode (GC/MS; Agilent 7890A/Agilent 5975C; inert XL MSD with triple axis detector). Isoprene was purged from the water sample with helium for 15 minutes and dried using a Nafion membrane dryer (Perma Pure; ASTRA-OMZ) or potassium

carbonate (SPACES/OASIS). Before being injected into the GC, isoprene was preconcentrated in a trap cooled with liquid nitrogen. Gravimetrically prepared liquid standards in ethylene glycol were measured in the same way as the samples and used to perform daily calibrations for quantification. Gaseous deuterated isoprene (isoprene-d5) was measured together with each sample as an internal standard to account for possible sensitivity drift between calibrations. The precision for isoprene measurements was ± 8%.

Air samples were collected in electropolished stainless steel flasks and pressurized to approximately 2.5 atm with a metal bellows pump. Analysis was conducted after samples were returned to the laboratory. Isoprene was measured along with a range of halocarbons, hydrocarbons, and other gases using a combined GC/MS/FID/ECD system with a modified Markes Unity II/CIA sample preconcentrator. The modifications incorporated a water removal system consisting of a cold trap (-20°C) and a Perma Pure drier (MD-050-24). Isoprene and >C4

hydrocarbons were quantified using selected ion MS and were calibrated against a whole air sample that is referenced to a NIST hydrocarbon mixture using GC/FID. Precision for isoprene is estimated at approximately ±0.4 ppt +5%.





### 3 Results and discussion

#### 3.1 Comparison of modeled and *in situ* measured isoprene data

The shipboard isoprene measurements from the ANT-XXV/1 cruise ranged from 2-157 pmol L$^{-1}$, with the highest levels in the subtropics of the southern hemisphere, and lower levels in the tropics (Figure 2). Model

simulations were carried out along the cruise track using monthly mean satellite data from November 2008 for chl-*a*, surface winds, SST, and MLD as input parameters. The simulations underestimated the measured isoprene concentrations significantly, by as much as a factor of 20 over most of the cruise track (mean error of 19.1 pmol L$^{-1}$). Simulations were also carried out using *in situ* shipboard measurements (chl-*a*, wind speed, SST, MLD) as the input parameters. In both cases, the model simulations show a peak in the calculated isoprene

levels at 13-17°N which is not present in the observations, whereas the peak, using *in situ* data as input parameter, is much smaller. This peak corresponds to elevated chl-*a* concentrations, suggesting that while there may have been high biological activity in this region, isoprene producing species were not abundant (Figure 3, 4). These results demonstrate that a single isoprene production factor and bulk chl-*a* concentration do not adequately describe the variability in isoprene production. When isoprene producing PFTs are dominant,

however, the modeled isoprene values follow the observed isoprene values (increasing isoprene concentration north of 33°N, Figure 2, 5). The elevated isoprene concentrations in the subtropics of the southern hemisphere are not represented by the model.

Monthly mean satellite data cannot resolve rapid changes like short phytoplankton blooms or wind events. We compared the satellite data to the ship's *in situ* measurements of SST, wind speed, calculated MLD, and *in situ*

measured chl-*a* concentration as input parameters for the model (Figure 2), in order to determine if the resolution of the satellite data does resolve important features. The modeled isoprene concentrations closely follow the variability in chl-*a*, demonstrating that chl-*a* has the strongest influence of the four input parameters to the model. The differences between modelled isoprene concentrations using *in situ* data vs satellite data are due primarily to the differences in chl-*a* (*in situ* data in general two times higher than satellite data) with the largest

differences in the regions from 10-25°N and 40-45°N. As the discrepancies between *in situ* and satellite data are significant, *in situ* measured data of chl-*a* are used from now on for further calculations with the ISO$_{PS05}$-model. Using monthly mean satellite data for wind speed, SST and climatological values for MLD does not bias the model results significantly relative to the *in situ* data.

#### 3.2 Modeling isoprene production using PFTs and revised $k_{BIOL}$

Palmer and Shaw (2005) used a universal P$_{chloro}$-value of 1.8 ± 0.7 µmoles (g chl-*a*)$^{-1}$ day$^{-1}$ based on laboratory phytoplankton monoculture experiments with several cyanobacteria, eukaryotes, and coccolithophores (Table 1; Shaw et al., 2003). Subsequent laboratory experiments with monocultures of different phytoplankton species have shown generally higher isoprene production rates with large variations between PFTs (Arnold et al., 2009;Bonsang et al., 2010;Colomb et al., 2008;Exton et al., 2013). In addition, Tran et al. (2013) observed that

isoprene concentrations in the field are highly PFT-dependent.

We averaged the P$_{chloro}$-values of different PFTs (Table 2) and multiplied these values by the amount of the corresponding PFT. Using PFTs instead of total biomass of phytoplankton (chl-*a*) in the model run results in higher isoprene model concentrations (orange, Figure 4), which match the overall isoprene concentration levels measured north of 10°N quite well. However, there are also regions where the model still cannot reproduce the




measured isoprene concentrations. Between 10°N and 25°S the calculated isoprene concentrations are quite stable with only small variations between 6 and 23 pmol L$^{-1}$. Measured concentrations are slightly higher between 10°N and 12°S (15-30 pmol L$^{-1}$), and sharply increase to 40-60 pmol L$^{-1}$ south of 12°S with a maximum concentration of 150 pmol L$^{-1}$ (16°S). As there were no significant differences in wind speed, SST or MLD in

these two regions during the cruise, there must be at least one additional source which is not captured in the model. In contrast, at 15°N and at 22°N the model overestimates the isoprene concentration (Figure 4). Chl-*a* concentrations are 10-20 times higher in these two areas than elsewhere on the cruise (Figure 3) and dominated by diatoms. However, the calculated isoprene is not 10-20 times higher, since diatoms have a relatively low P$_{chloro}$-value (2.54 µmol (g chl-*a*)$^{-1}$ day$^{-1}$) and, therefore, using their respective PFT value modulates the influence

of the increased chl-*a* on isoprene concentrations (Figure 5).

Excluding the two bloom areas, the main PFTs contributing to the modeled isoprene concentrations were prokaryotic phytoplankton (cyanobacteria and *Prochlorococcus*) and haptophytes (Figure 5, see also Taylor et al., 2011). It should be noted that the PFTs considered in our study are only part of the full phytoplankton community. In addition, these values can be easily over- or underestimated, due to a high variability in the P$_{chloro}$-

values within one group of PFTs (e.g. haptophytes: 1 - 15.36 µmol isoprene (g chl-*a*)$^{-1}$ day$^{-1}$; Table 2). Using the ISO$_{PFT-kBIO}$ model approach the isoprene concentrations increase by a factor of 1.35, resulting in better agreement with the observations (Figure 4). Overall for the conditions of this cruise, the ISO$_{PFT-kBIO}$ model simulation yields 12-fold higher isoprene levels than ISO$_{PS05}$ (mean error of 11.8 pmol L$^{-1}$).

It is obvious that even after implementing these changes the model does not reproduce all the measured isoprene

values or their distribution pattern. One particular problem is that marine isoprene emissions are very low in comparison to terrestrial isoprene emissions. Coastal emissions have to be calculated and interpreted carefully due to this terrestrial influence. We assume no terrestrial influence in the open ocean, since the atmospheric lifetime of isoprene is short. Despite the terrestrial influence on atmospheric isoprene values over the ocean, calculating surface ocean isoprene concentrations, other assumptions in the model should be scrutinized in order

to understand the discrepancies between measured and calculated values:

1) The model assumes well mixed isoprene concentrations through the MLD, which is, in fact, not the case. Measurements of depth profiles show a vertical gradient with a maximum of isoprene at the depth of the chl-*a* maximum slightly below the MLD (Bonsang et al., 1992;Milne et al., 1995;Moore and Wang, 2006), which was also measured during our three campaigns (data not shown). Gantt et al. (2009) tried to solve this

problem using a light dependent isoprene production rate, but this resulted in high fluxes in the tropics that are questionable when compared to field measurements.

2) Using PFT dependent production rates strongly improved the model by adding more specific and realistic product information. Nonetheless, we may still be missing some important species within the PFTs and the average taken over the isoprene measurements among the cultured species within one PFT carries some

uncertainty. We used up to eight different PFTs, illustrating that only the four main groups (haptophytes, cyanobacteria, *Prochlorococcus* and diatoms) produce the most isoprene (Figure 5). These groups are also the only four detected by the satellite product PHYSAT (Alvain et al., 2005), which has been used previously for predictions of isoprene (Arnold et al., 2009;Gantt et al., 2009). However, neglecting the other PFTs might lead to different results (others, Figure 5). This highlights the need to measure the isoprene

emission of more species within each PFT group under different physiological conditions. This would also,





potentially, lower the uncertainty of global marine isoprene emissions, which was found to be in the range of 20% when using the upper or lower bounds of PFT dependent production rates (Gantt et al., 2009).

3) The temporal resolution of the simple model may also not be adequate. Gantt et al. (2009) could show that their model using remote sensing input in combination with the light dependence of isoprene production overestimated daytime isoprene concentrations and underestimated nighttime concentrations compared to the high temporal resolution field measurements of Matsunaga et al. (2002). The possible diurnal cycle of isoprene could not be resolved with remote sensing data obtained only at a specific local time during the day (e.g. 10 am for MODIS-Terra and 1 pm for MODIS-Aqua).

4) The role of bacteria in producing isoprene is also unclear and may be a missing variable in the steady state equation. Alvarez et al. (2009) observed bacterial isoprene production in estuary sediments and discovered isoprene production using different cultures of bacteria. However, Shaw et al. (2003) could not find any evidence of bacterial isoprene production in separate experiments.

### 3.3 Verification of the ISO$_{PFT-kBIO}$ model using data from the Indian and east Pacific Oceans

Isoprene concentrations calculated with the original (ISO$_{PS05}$) and revised (ISO$_{PFT-kBIO}$) model are compared to measured isoprene in the surface ocean at two additional campaigns in two widely differing ocean basins (Indian Ocean, SPACES/OASIS, 2014; East Pacific Ocean, ASTRA-OMZ, 2015). The original model ISO$_{PS05}$ predicts on average 19±12 times lower isoprene concentrations compared with measured values for the additional two ship campaigns (circles, Figure 6), which confirms the results obtained for ANT-XXV/1. With the newly determined (lower) value for $k_{BIOL}$ and PFT dependent P$_{chloro}$-values, the ISO$_{PFT-kBIO}$ model predicts concentrations that are 10 times higher than the original model ISO$_{PS05}$ output (crosses, Figure 6). This leads to a mean underestimation of 1.7±1.2 between modeled and measured isoprene concentrations. The main cause of the better agreement between measured and modeled isoprene concentrations is the isoprene production rate related to the production input parameter (color coding, Figure 6). The mean isoprene production rate using chl-$a$ as input parameter multiplied by a factor of 1.8 µmol (g chl-$a$)$^{-1}$ day$^{-1}$ is less than 0.5 pmol L$^{-1}$ day$^{-1}$, which is insufficient to explain the measured concentrations in all three campaigns. Using P$_{chloro}$-values multiplied with the concentration of the related PFT yields in an isoprene production rate of 1-2 pmol L$^{-1}$ day$^{-1}$ in non-bloom areas and even higher rates during phytoplankton blooms, resulting in isoprene concentrations that are comparable to the measured ones. The opposite can also occur, as seen on DOY 322 (Figure 6), when PFT specific production rates are smaller than those using chl-$a$ only, due to the dominance of a low isoprene producing PFT.

### 4 Global oceanic isoprene emissions and implications for marine aerosol formation

Monthly mean global ocean isoprene concentrations were calculated using the revised model ISO$_{PFT-kBIO}$ (2°x2° grid). As there were no PFT satellite data readily available, we used an empirical relationship between chl-$a$ and PFTs as parameterized by Hirata et al. (2011). The quality of this parameterization was verified against the PFT datasets from all three campaigns (discrepancy less than 25%) and is shown in Figure 6 (grey diamonds). Monthly mean global ocean isoprene emissions (Figure S1-S12, supplement) were averaged in order to compute global sea-to-air fluxes of isoprene for 2014 (Figure 7). An annual emission of 0.21 Tg C was calculated, which is two times higher than the value estimated by Palmer and Shaw (2005) (0.11 Tg C yr$^{-1}$). The highest emissions,




more than 100 nmol m$^{-2}$ day$^{-1}$, can be seen in the North Atlantic Ocean and the Southern Ocean, associated with high biological productivity and strong winds driving the air-sea gas exchange. The influence of regional hot spots of biological productivity, such as the upwelling off Mauretania or the Brazil-Malvinas Confluence Zone, can also be seen. The tropics (23.5°S-23.5°N) account for only 28% of global isoprene emissions, but represent

~47% of the world oceans.

Yearly emissions of 0.21 Tg C are at the lower end of the range of previously published studies (Arnold et al. (2009) 0.27 Tg C yr$^{-1}$; Gantt et al. (2009) 0.92 Tg C yr$^{-1}$). Both studies use remotely sensed PFT data instead of chl-*a* to evaluate the isoprene production. Unlike this study, they implemented the Alvain et al. (2005) approach using PHYSAT data, which uses spectral information to produce global distributions of the dominant PFT, but is

limited to four phytoplankton groups (haptophytes, *Prochlorococcus*, *Synechococcus* and diatoms). It should be noted that PHYSAT does not provide actual concentrations, but rather only the relative dominance of the four groups. Arnold et al. (2009) used similar assumptions as Palmer and Shaw (2005) to calculate isoprene loss, namely that loss in the water column by advective mixing and aqueous oxidation is on a longer timescale than loss by air-sea gas exchange and, therefore, negligible. Thus, their calculated emissions of 0.27 Tg C yr$^{-1}$ are an

upper estimate. The approach of Gantt et al. (2009) had two main differences compared to our study: 1) Instead of using the MLD climatology of de Boyer Montégut et al. (2004), they used a maximum depth where isoprene production can occur as calculated by the downwelling irradiance (using the diffuse attenuation coefficient values at 490 nm) and the light propagation throughout the water column that is estimated by using Lambert-Beer's Law. 2) They tested two of the detectable PFTs in laboratory experiments using monocultures of diatoms

and coccolithophores growing under different light conditions to evaluate light intensity dependent isoprene production rates. Light intensity dependent production rates of *Prochlorococcus* and *Synechococcus* were derived after Gantt et al. (2009) using the original production rates at a specified wavelength measured by Shaw et al. (2003). Their isoprene emission calculations are more than four times higher than calculated with our approach, probably as a result of the light dependent isoprene production rates. Whereas our global map shows

very low emissions in the tropics due to a low phytoplankton productivity, the emissions modelled by Gantt et al. (2009) are comparable to those of high productivity areas like the Southern Ocean or the North Atlantic Ocean, likely as a consequence of the high solar radiation in the tropics. The data from our three cruises contradict this model-derived result, and show very low concentrations in the tropical regions, which implies a very low flux of isoprene to the atmosphere. Furthermore, Meskhidze et al. (2015) showed that at a specific light intensity the

isoprene production rate of tested monocultures sharply decreases.

Using atmospheric isoprene concentrations measured in two of three campaigns, we were able to use a top-down approach to calculate isoprene emissions in order to compare with the bottom-up flux estimates. We used a box model with an assumed marine boundary layer height of 800 m, which reflected the local conditions during the two campaigns. The only source of isoprene for the air was assumed to be the sea-to-air flux (emission) and the

atmospheric lifetime was assumed to be determined by reaction with OH (chemical loss). The concentration outside the box was assumed to be the same as inside to neglect advection in to and out of the box. The resulting steady-state concentration in the box, as a function of flux and lifetime, is shown in Figure 8 (for a one hour lifetime it takes approximately 10 hours to achieve steady state). For comparison, the mean measured concentration of isoprene in the atmosphere during the two cruises (2.5 ± 1.5 ppt) is also plotted in Figure 8. The

measured concentrations match previously measured remote open ocean atmospheric values (Shaw et al., 2010). We only used atmospheric measurements which were obtained during daytime (to reflect reaction with OH) and





which were not influenced by terrestrial sources. This was determined by omitting data points with concomitant high levels of anthropogenic hydrocarbons (concentrations of butane higher 20 ppt). Reported mean atmospheric lifetime estimates of isoprene range from minutes up to four hours, depending mainly on the atmospheric concentration of OH (Pfister et al., 2008). We calculate that for a conservatively estimated lifetime of 4 h, a sea-

to-air flux of at least 200 nmol m$^{-2}$ day$^{-1}$ is needed to reach the lower range of the atmospheric concentration measured during SPACES/OASIS and ASTRA-OMZ, which is approximately 10 times higher than computed (Figure 7, 8). Recent studies showed that the measured fluxes of isoprene range from 4.6-148 nmol m$^{-2}$ day$^{-1}$ in June/July 2010 in the Arctic (Tran et al., 2013) to 181.0-313.1 nmol m$^{-2}$ day$^{-1}$ in the productive Southern Ocean during austral summer 2010/2011 (Kameyama et al., 2014). Despite these high literature values, it appears that

the calculated fluxes cannot explain the measured atmospheric concentrations when a lifetime of 4 h is assumed.

**5 Conclusions**

The revised Palmer and Shaw (2005) isoprene emission model was evaluated against direct surface ocean isoprene measurements from three different ocean basins, yielding comparable ocean concentrations that were slightly underestimated (factor of 1.7±1.2). The resulting annual, global oceanic isoprene emissions are two

times higher than the calculated flux with the original model. However, using a simple top-down approach based on measured atmospheric isoprene levels, we calculate that emissions from the ocean are required to be more than one order of magnitude greater than those computed using the bottom-up estimate based on measured oceanic isoprene levels. This result is consistent with a numerical evaluation of global ocean isoprene emissions by Luo and Yu (2010). One possible explanation could be production in the surface microlayer (SML) that is not

simulated by the model. Ciuraru et al. (2015) showed that isoprene is produced photochemically by surfactants in an organic monolayer at the air-sea interface. As the SML is enriched with surfactants (Wurl et al., 2011), the isoprene flux from the SML could range from 1000-33000 nmol m$^{-2}$ day$^{-1}$, which is much larger (about 2 orders of magnitude) than the highest fluxes calculated from our observations. To date there is no evidence of such a large gradient in the surface ocean between the surface and 10 m. Thus, further field measurements probing the

SML could be a step forward in reconciling the role of the ocean for the atmospheric isoprene budget. Using the bottom-up approach, isoprene emissions are much smaller and given this scenario, isoprene consequently appears to be a relatively insignificant source of OC in the remote marine atmosphere. Arnold et al. (2009) calculated a yield of 0.04 Tg yr$^{-1}$ OC derived from marine isoprene by using yearly emissions of 1.9 Tg yr$^{-1}$ and a SOA yield of 2% (Henze and Seinfeld, 2006). This is equivalent to 0.5% of estimated 8 Tg yr$^{-1}$ global source

of oceanic OC (Spracklen et al., 2008). Using our bottom-up emission of 0.21 Tg C yr$^{-1}$ will even lower this small influence. Until this discrepancy between bottom-up and top-down approaches is resolved, the question of whether isoprene is a main precursor to remote marine boundary layer particle formation still remains open.

**Acknowledgements**

The authors would like to thank the captain and crew of the R/V Polarstern (ANT-XXV/1) and R/V Sonne

(SPACES/OASIS and ASTRA-OMZ) as well as the chief scientists, Gerhard Kattner (ANT-XXV/1) and Kirstin Krüger (SPACES/OASIS). Boris Koch and Birgit Quack were also provided valuable help. We thank Sonja Wiegmann and Wee Cheah for pigment lab work during SPACES/OASIS and HPLC pigment analysis of SPACES/OASIS and ASTRA-OMZ samples and Rüdiger Röttgers for helping with the pigment sampling





during ASTRA-OMZ. Paul I. Palmer gratefully acknowledges his Royal Society Wolfson Research Merit Award. Elliot Atlas acknowledges support from the NASA UARP program and thanks Leslie Pope and Xiaorong Zhu for assistance in canister preparation. The authors gratefully acknowledge the NOAA Air Resources Laboratory (ARL) for the provision of the HYSPLIT transport and dispersion model used in this

publication as well as NASA for providing the satellite MODIS-Aqua and MODIS-Terra data. QuikScat and SeaWinds data were produced by Remote Sensing Systems with thanks to the NASA Ocean Vector Winds Science Team for funding and support. This work was carried out under the Helmholtz Young Investigator Group of Christa A. Marandino, TRASE-EC (VH-NG-819), from the Helmholtz Association through the President's Initiative and Networking Fund and the GEOMAR Helmholtz-Zentrum für Ozeanforschung Kiel.

The R/V Sonne cruises SPACES/OASIS and ASTRA-OMZ were financed by the BMBF through grants 03G0235A and 03G0243A, respectively.

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

**Table 1: List of parameters used in each model.**

| Parameter | Abbreviation | Unit | Model approach | | |
|---|---|---|---|---|---|
| | | | $ISO_{PS05}$ | $ISO_{PFT}$ | $ISO_{PFT-kBIO}$ |
| Isoprene production rate | $P$ | pmol L$^{-1}$ day$^{-1}$ | $P_{chloro}*$[chl-$a$] | $P_{chloro}*$[PFT] | $P_{chloro}*$[PFT] |
| Chemical loss rate | $k_{OH}*C_{OH}$ | day$^{-1}$ | 0.0518 | 0.0518 | 0.0518 |
| | $k_{O2}*C_{O2}$ | day$^{-1}$ | 0.0009 | 0.0009 | 0.0009 |
| Biological loss rate | $k_{BIOL}$ | day$^{-1}$ | 0.06 | 0.06 | 0.01 |
| Gas transfer coefficient | $k_{AS}$ | m s$^{-1}$ | Wanninkhof (1992) | | |
| Mixed layer depth | MLD | m | de Boyer Montégut et al. (2004) | | |
| Mixing loss rate | $L_{MIX}$ | pmol L$^{-1}$ day$^{-1}$ | 0.0459 | 0.0459 | 0.0459 |
| Chl-$a$ normalized isoprene production rate | $P_{chloro}$ | μmol (g chl-$a$)$^{-1}$ day$^{-1}$ | 1.8 | PFT dependent (Table 2) | |

**Table 2: Chlorophyll-normalized isoprene production rates ($P_{chloro}$) determined from analysis of phytoplankton cultures experiments described in the literature (Exton et al. (2013) and references therein). $P_{chloro}$-values are given in μmol (g chl-$a$)$^{-1}$ day$^{-1}$.**

| Species | Literature $P_{chloro}$-value | Averaged $P_{chloro}$-values for specific PFTs | References |
|---|---|---|---|
| **Bacillariophyceae** | | | |
| *Chaetoceros neogracilis* (CCMP1318) | 28.48 | | Colomb et al. 2008 |
| *Cheatoceros neogracilis* (CCMP 1318) | 1.26 ±1.19 | | Bonsang et al. 2010 |
| *Thalassiosira pseudonana* (CCAP 1085/12) | 5.76 ±0.24 | 2.54 | Exton et al. 2013 |
| *Pelagomonas calceolate* (CCMP 1214) | 1.6 ±1.6 | | Shaw et al. 2003 |
| *Phaeodactylum tricornutum* (Falkowski) | 2.85 | | Colomb et al. 2008 |


| | | | |
|---|---|---|---|
| *Phaeodactylum tricornutum* (UTEX646) | 1.12 ±0.32 | | Bonsang et al. 2010 |
| *Skeletonema costatum* | 1.32 ±1.21 | | Bonsang et al. 2010 |
| *Skeletonema costatum* (CCMP 1332) | 1.8 | | Shaw et al. 2003 |
| *Thalassiosira weissflogii* (CCMP 1051) | 4.56 ±0.24 | | Exton et al. 2013 |
| Diatoms (elsewhere) | 2.48 ±1.75 | | Arnold et al. 2009 |
| *Cylindrotheca sp.* | 2.64 | | Exton et al. 2013 |
| | | | |
| **cold adapted Bacillariophyceae** | | | |
| *Fragilariopsis kerguelensis* | 0.56 ±0.35 | | Bonsang et al. 2010 |
| *Chaetoceros debilis* | 0.65 ±0.2 | | Bonsang et al. 2010 |
| *Chaetoceros muelleri* (CCAP 1010/3) | 9.36 ±1.2 | Excluded from the | Exton et al. 2013 |
| *Fragilariopsis cylindrus* | 0.96 ±0.24 | average isoprene | Exton et al. 2013 |
| *Nitzschia sp.* (CCMP 1088) | 0.96 ±0.24 | production rate | Exton et al. 2013 |
| *Synedropsis sp.* (CCM 2745) | 0.72 ±0.24 | | Exton et al. 2013 |
| Diatoms (Southern Ocean) | 1.21 ±0.57 | | Arnold et al. 2009 |
| | | | |
| **Dinophyceae** | | | |
| *Prorocentrum minimum* | 10.08 ±1.44 | | Exton et al. 2013 |
| *Symbiodinium sp.* (CCMP 2464) | 4.56 ±3.12 | | Exton et al. 2013 |
| *Symbiodinium sp.* (CCMP 2469) | 17.04 ±8.4 | 13.78 | Exton et al. 2013 |
| *Symbiodinium sp.* | 9.6 ±2.8 | | Exton et al. 2013 |
| *Symbiodinium sp.* (CCMP 2463) | 27.6 ±1.68 | | Exton et al. 2013 |
| | | | |
| **Cyanophyceae** | | | |
| *Prochlorococcus sp.* (axenic MED4) (high light) | 1.5 ±0.9 | 1.5 | Shaw et al. 2003 |
| *Prochlorococcus* | 9.66 ±5.78 | 9.66 | Arnold et al. 2009 |
| *Synechococcus sp.* (RCC 40) | 4.97 ±2.87 | | Bonsang et al. 2010 |
| *Synechococcus sp.* (WH 8103) | 1.4 | 6.04 | Shaw et al. 2003 |
| *Synechococcus sp.* (CCMP 1334) | 11.76 ±0 | | Exton et al. 2013 |
| | | | |
| **Chlorophyceae** | | | |
| *Dunaliella tertiolecta* | 0.36 ±0.22 | | Bonsang et al. 2010 |
| *Dunaliella tertiolecta* (DUN, Falkowski) | 2.85 | 1.47 | Colomb et al. 2008 |
| *Dunaliella tertiolecta* (CCMP 1320) | 1.2 | | Exton et al. 2013 |
| | | | |
| **Cryptophyceae** | | | |
| *Rhodomonas lacustris* (CCAP 995/3) | 9.36 ±0.72 | 9.36 | Exton et al. 2013 |
| | | | |
| **Prasinophyceae** | | | |
| *Micromonas pusilla* (CCMP 489) | 1.4 ±0.8 | | Shaw et al. 2003 |
| *Prasinococcus capsulatus* (CCMP 1614) | 32.16 ±5.76 | 12.47 | Exton et al. 2013 |
| *Tetraselmis sp.* (CCMP 965) | 3.84 ±0.24 | | Exton et al. 2013 |
| | | | |
| **Prymnesiophyceae** | | | |
| *Calcidiscus leptoporus* (AC365) | 5.4 | | Colomb et al. 2008 |
| *Emiliania huxleyi* (CCMP 371) | 11.54 | 6.92 | Colomb et al. 2008 |
| *Emiliania huxleyi* (CCMP 371) | 1 | | Bonsang et al. 2010 |



| | | |
|---|---|---|
| *Emiliania huxleyi* (CCMP 373) | 1 ±0.5 | Shaw et al. 2003 |
| *Emiliania huxleyi* (CCMP 373) | 2.88 ±0.48 | Exton et al. 2013 |
| *Emiliania huxleyi* (CCMP 1516) | 11.28 ±0.96 | Exton et al. 2013 |
| *Gephyrocapsa oceanica* | 15.36 ±4.1 | Exton et al. 2013 |

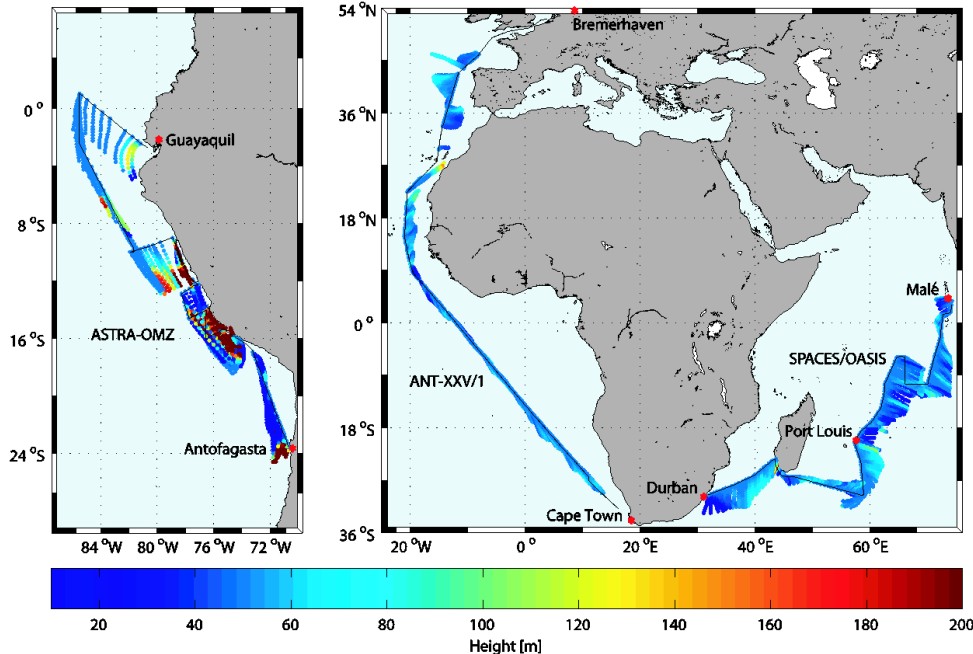

**Figure 1: Cruise tracks (black) of ANT-XXV/1 (November 2008, East Atlantic Ocean), SPACES/OASIS (June/July 2014, Indian Ocean) and ASTRA-OMZ (October 2015, East Pacific Ocean). Air mass back trajectories calculated for 12 hours with a starting height of 50 m using HYSPLIT are superimposed on the cruise track. Color coding indicates altitude about sea level.**





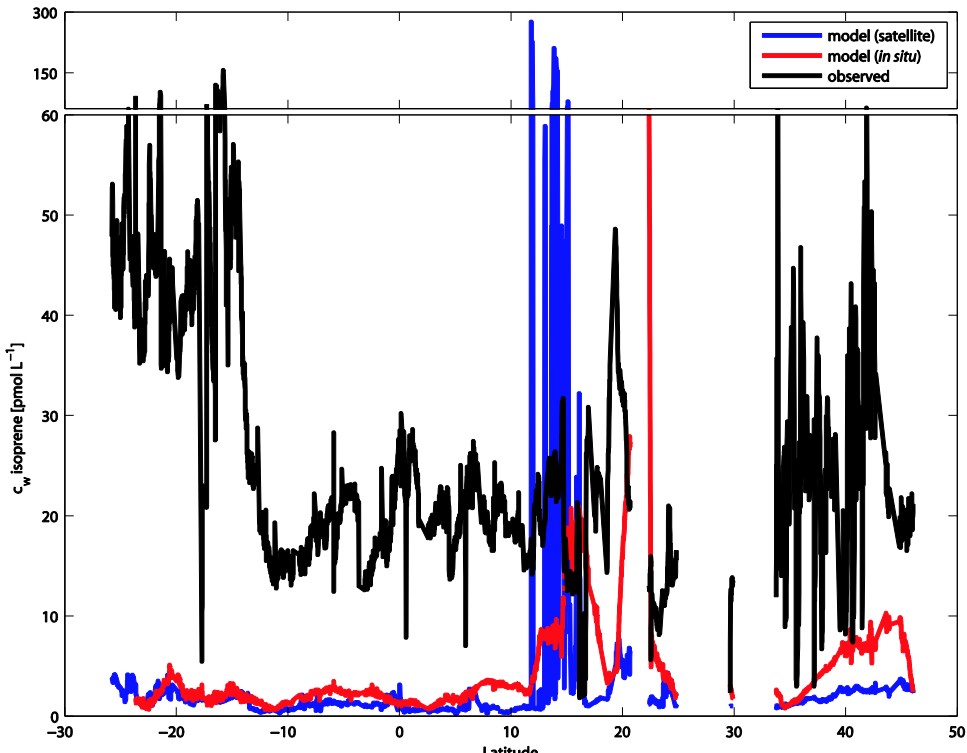

**Figure 2: Comparison of observed (black) and modeled seawater isoprene concentrations for the ANT-XXV/1 cruise. Model calculations were carried out using the ISO$_{PS05}$ model configuration, with monthly mean satellite data (blue) for chl-$a$, wind speed, SST, and MLD (climatology) and $in$ $situ$ shipboard measurements (red).**





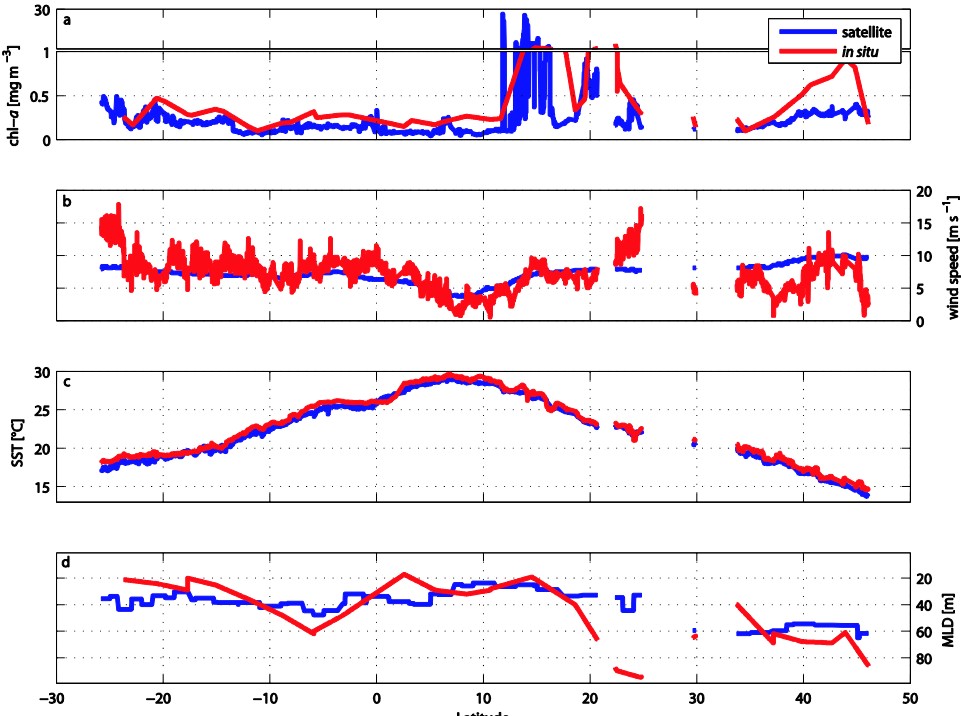

**Figure 3: Satellite and *in situ* data for the ANT-XXV/1 cruise. Monthly mean satellite derived data (blue) and *in situ* measurements (red) of (a) chl-*a*, (b) wind speed, (c) SST. (d) Monthly mean climatology values (blue) and in situ measurements (red) of MLD.**




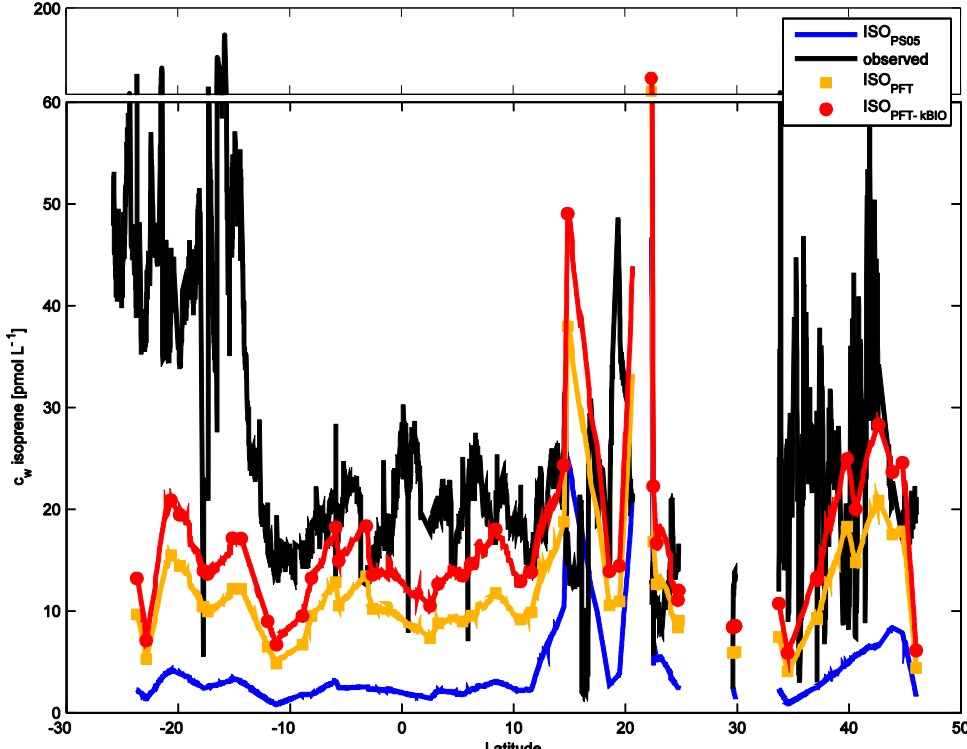

**Figure 4: Comparison of** *in situ* **measured isoprene (black) with model derived isoprene concentrations for the ANT-XXV/1 cruise using ISO$_{PS05}$ (blue), ISO$_{PFT}$ (orange) and ISO$_{PFT-kBIO}$ (red); squares and circles: direct measurements; solid lines: interpolated data.**





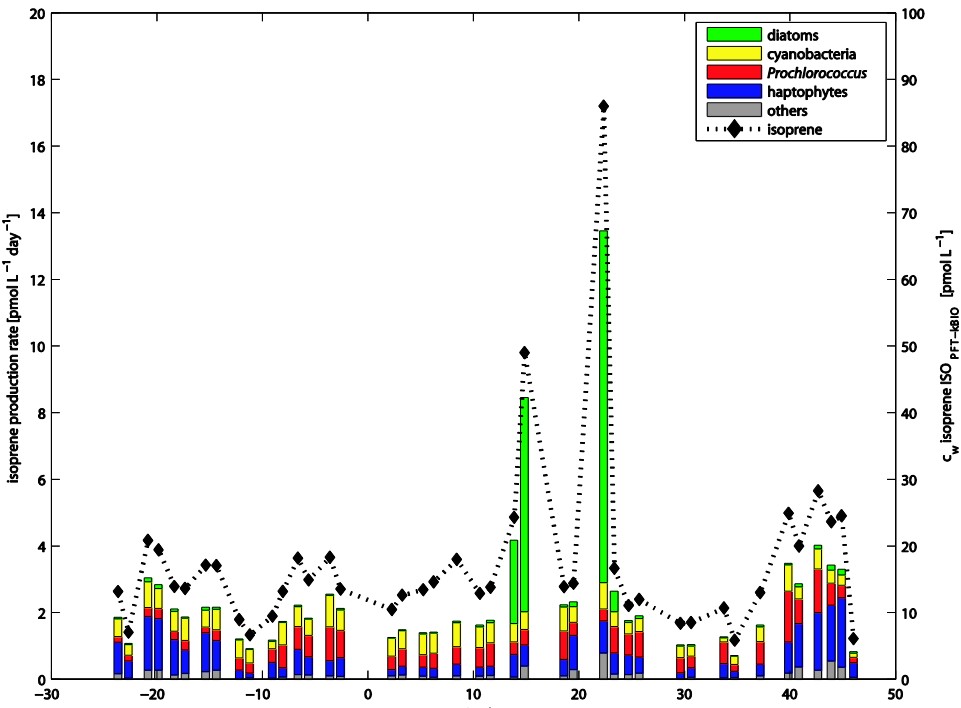

Figure 5: Proportion of main PFTs contributing to the total isoprene production rate for each station during ANT-XXV/1.





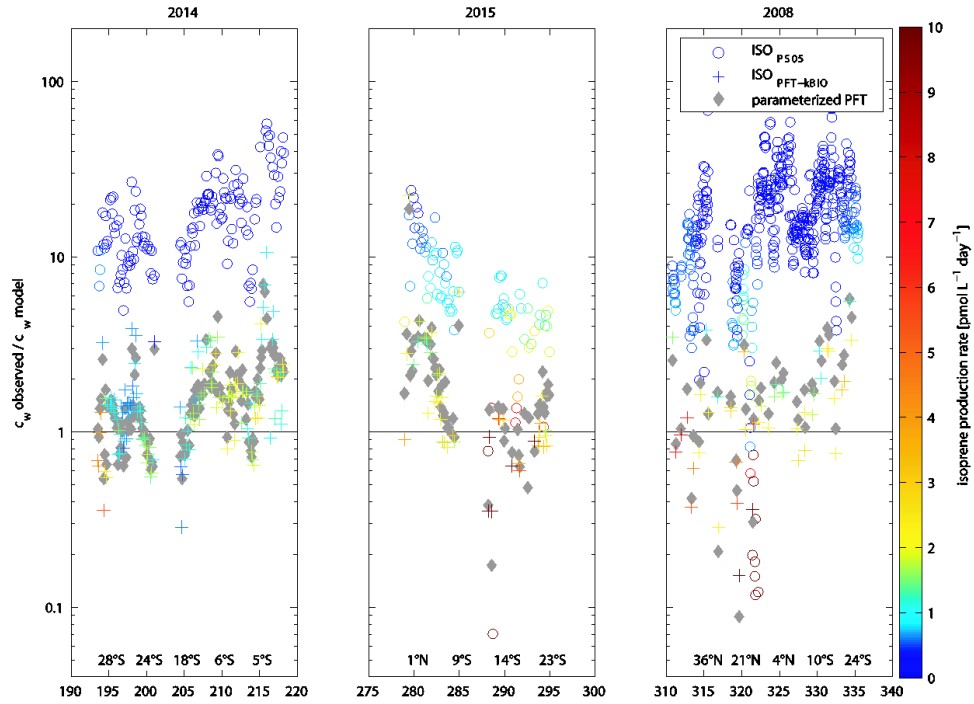

**Figure 6: Observed isoprene concentration divided by modeled isoprene concentration on a logarithmic scale for three different cruises; left: SPACES/OASIS 2014, middle: ASTRA-OMZ 2015, right: ANT-XXV/1 2008; circles and crosses represent data derived by the original ISO$_{PS05}$ and revised ISO$_{PFT-kBIO}$ model, respectively; every data point is color coded with the corresponding isoprene production rate input parameter; grey diamonds represent data using parameterized PFT data by Hirata et al. (2011); the black line represents a ratio of 1.**





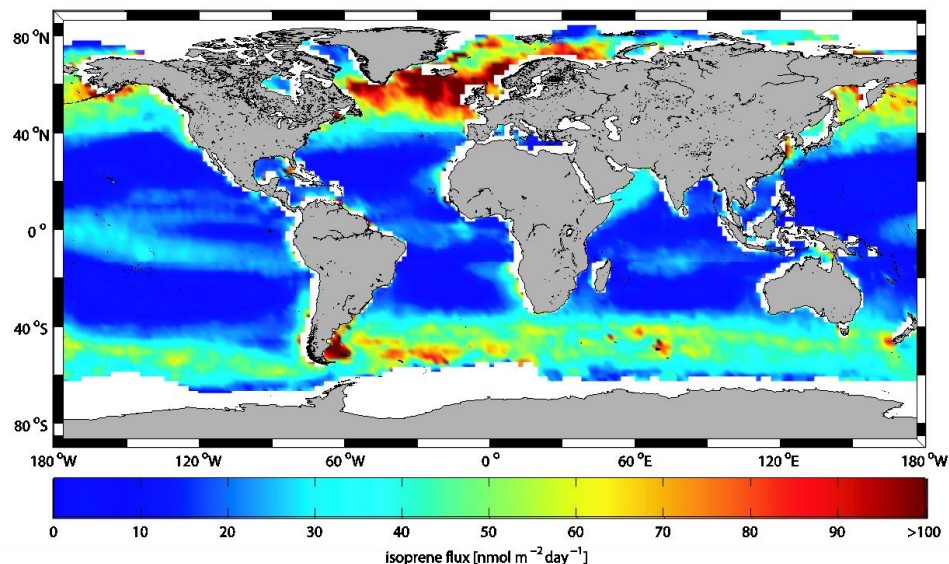

Figure 7: Global marine isoprene fluxes in nmol m$^{-2}$ day$^{-1}$ for 2014.





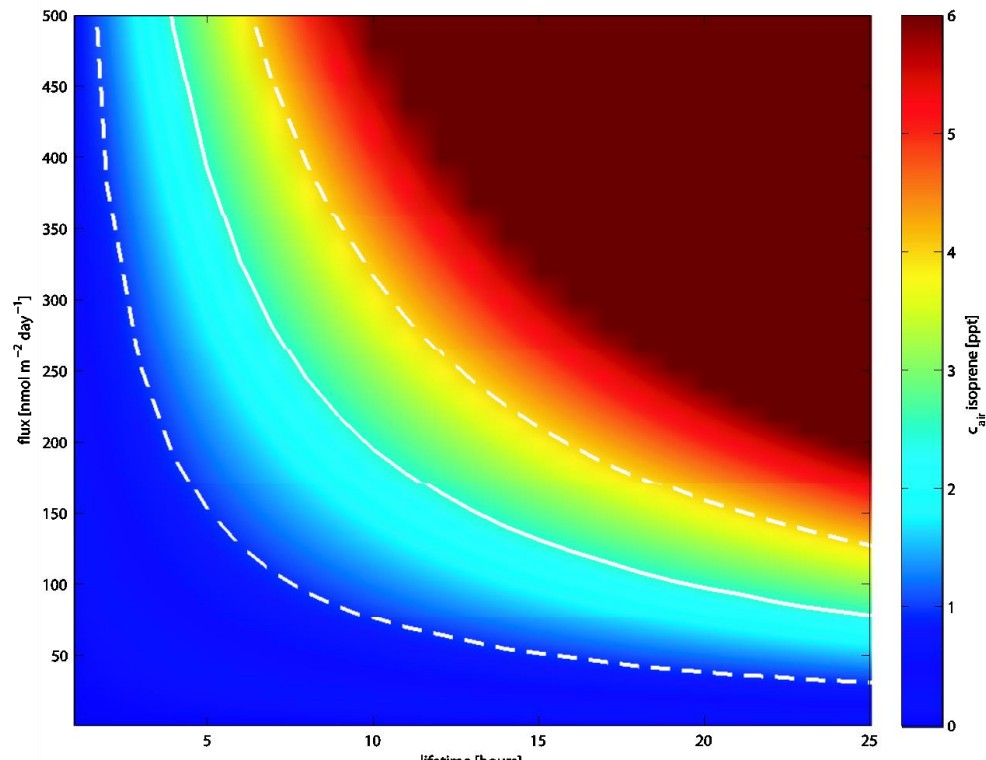

**Figure 8: Daytime isoprene mixing ratios (ppt) in a marine atmospheric boundary layer of 800m height as a function of the sea-to-air flux and the atmospheric lifetime based on a simple box model approach; solid white line reflects the mean air values of isoprene during SPACES/OASIS and ASTRA-OMZ; The dashed lines represent one standard deviation from the mean.**