# Peer review of "Can simple models predict large scale surface ocean isoprene concentrations?"

_Atmospheric Chemistry and Physics, 2016_

## Referee Comment (RC1) · Anonymous Referee #2 · 19 Jul 2016

General Comments: This manuscript describes an evaluation of the Palmer and Shaw, model (which parameterized oceanic isoprene concentrations as a function of chlorophyll concentrations and laboratory isoprene emission factors) with satellite chlorophyll data and in situ ocean cruise measurements. The Booge et al. manuscript then describes subsequent updates and extensions of that model, with evaluations based on the cruise data. The updates included (1) the addition of emission factors representing multiple individual phytoplankton functional types (PFTs) as opposed to a single average value across PFTs, (2) the testing of the model results against individual pigment markers), and (3) laboratory measurement of biological degradation rate with an isotopically labeled isoprene and subsequent inclusion into the model. The results demonstrated large increases in predicted oceanic concentrations, and thus fluxes, which more closely matched the in situ cruise data than the original model. However,

the fluxes are still insufficiently high to match observed atmospheric isoprene concentrations. The authors conclude missing sources of oceanic isoprene still exist.

This is a very well-performed study which has successfully updated the prior model, which was limited by necessity to representing only a few phytoplankton species and functional types. In the intervening years, a number of laboratory studies have been performed with dozens of additional phytoplankton species and several additional PFTs, thus broadly expanding the information available with which to expand the model. The experiment to make direct measurements of isoprene biological uptake through deuteraetd material is particularly exciting, and it will be important to follow through with that expected publication as indicated.

Booge et al. do a remarkable job at combining all the new data sources and model formulation. The results have increased oceanic concentration predictions substantially, which partially compensates for previously-expected "missing sources", but these are still only of marginal importance to the air concentration underestimate. Additionally, there are clear locations during the cruises where the updated model still fails to reproduce the appropriate concentrations. Despite the fact this mismatch between bottom-up (parameterized fluxes and concentrations) and measured air concentrations still exists, this paper has performed important work, is a major step forward, and needs to be published. It is an important paper to the fields of remote chemistry, and aerosol formation marine regions.

The reporting is descriptive, succinct, and easy to follow. The analytical and measurement methodologies used are all robust and generally have been previously well-proven. The analyses are performed to an appropriate level of detail, the conclusions drawn are well-supported, and the literature is comprehensively cited.

Therefore, I recommend publication with minor revision, and have only minor comments below.

Specific Comments: Page 6, line 15 – If any species identifications beyond PFT identifcation by pigment (i.e. Figure 5) it would be helpful to point out whether they were species previously tested for isoprene production and present in Table 2 or not. This is particularly important in the areas where isoprene was not reproduced well.

Figure 6, line 20 – This should be Figure 3, not 2

Page 7, line 40 – I agree the physiological conditions can be a major driver of emission rates. A review of the laboratory studies that investigated this issue show a large range of emissions. This subject is worth a brief review of the relevant literature (∼2 sentences).

It is important to provide the caveat that the in-situ data provided is focused on three cruises in two regions of the oceans. It is a good test, but there are many regions that have not yet been tested with the updated model.

There is mention of the time resolution of the assessment being insufficient to capture the phytoplankton and isoprene heterogeneity that can result in large blooms of isoprene-producing species, and thus contributing to the underestimate of air concentrations. A sensitivity study based on bi-weekly or weekly satellite assessments of chlorophyll, as compared to monthly, would be an interesting addition to the manuscript. While it may not be possible to obtain MLD data on these time scales, perhaps there are pigment dat. Reasonable assumptions could be made in a simplistic manner to check what the maximum relative increase possible is in oceanic concentrations, flux, and ambient air concentrations. This would help determine if resolution is really the issue, or if untested high-producing species are the dominant cause of underestimate.

---

## Referee Comment (RC2) · Anonymous Referee #1 · 29 Jul 2016

General Comments: This article describes an evaluation of surface ocean isoprene predictions from a steady-state model using an extensive dataset of cruise data, remotely sensed satellite data, and box modeling. Although the topic of marine isoprene production isn't new, this work describes the most comprehensive evaluation to date using data from cruises spanning multiple years and oceanic regions. The methods are clearly described in the study, and figures effectively summarize the results. Beside some minor technical edits, the manuscript is very well written. My main critique of the article is the unevenness of the results; the oceanic concentrations are thoroughly evaluated while the discussion of the box modeling results are brief and overly suggestive. I'd suggest that the article be published after addressing the comments below.

Specific Comments: 1) After an comprehensive evaluation of the seawater isoprene

[Figure]

concentrations from the various cruises which clearly indicates the model improvement from the inclusion of phytoplankton functional types and reduction in bacterial degradation, I found the box modeling section of the results lacking. The article describes the existence of measured isoprene concentrations in the atmosphere from at least two cruises, yet these measurements are simply averaged and put into three curves of a figure. From this simplified analysis, an important conclusion is drawn (there are missing oceanic sources of isoprene) that appears in the abstract and conclusion of the article. I'd suggest either this analysis be removed or preferably expanded to include an evaluation of the atmospheric isoprene concentrations along the ship tracks. Specifically, I think the study could be informed by a box modeling study that moves with the ship location in order to identify the temporal and spatial extent of any missing oceanic isoprene sources.

2) The study clearly shows that phytoplankton function types can affect seawater isoprene concentrations, yet a comparison of measured and satellite-derived phytoplankton function type is not well described. I'd suggest describing in more detail the meaning of "discrepancy less than 25%" (Page 8, Line 35) in terms of the different phytoplankton functional types and oceanic regions and how any of these discrepancies may affect the uncertainty in the global marine isoprene emission estimate.

Minor Comments: 1) Page 1, Line 39: the yr-1 needs a superscript 2) Page 4, Line 19: should be "Table 2 of Taylor..."

———————————————————————

---

## Author Comment (AC1) · 26 Aug 2016

*General Comments: This manuscript describes an evaluation of the Palmer and Shaw, model (which parameterized oceanic isoprene concentrations as a function of chlorophyll concentrations and laboratory isoprene emission factors) with satellite chlorophyll data and in situ ocean cruise measurements. The Booge et al. manuscript then describes subsequent updates and extensions of that model, with evaluations based on the cruise data. The updates included (1) the addition of emission factors representing multiple individual phytoplankton functional types (PFTs) as opposed to a single average value across PFTs, (2) the testing of the model results against individual pigment markers), and (3) laboratory measurement of biological degradation rate with an isotopically labeled isoprene and subsequent inclusion into the model. The results demonstrated large increases in predicted oceanic concentrations, and thus fluxes, which more closely matched the in situ cruise data than the original model. However, the fluxes are still insufficiently high to match observed atmospheric isoprene concentrations. The authors conclude missing sources of oceanic isoprene still exist.*

*This is a very well-performed study which has successfully updated the prior model, which was limited by necessity to representing only a few phytoplankton species and functional types. In the intervening years, a number of laboratory studies have been performed with dozens of additional phytoplankton species and several additional PFTs, thus broadly expanding the information available with which to expand the model. The experiment to make direct measurements of isoprene biological uptake through deuteraetd material is particularly exciting, and it will be important to follow through with that expected publication as indicated.*

*Booge et al. do a remarkable job at combining all the new data sources and model formulation. The results have increased oceanic concentration predictions substantially, which partially compensates for previously-expected "missing sources", but these are still only of marginal importance to the air concentration underestimate. Additionally, there are clear locations during the cruises where the updated model still fails to reproduce the appropriate concentrations. Despite the fact this mismatch between bottomup (parameterized fluxes and concentrations) and measured air concentrations still exists, this paper has performed important work, is a major step forward, and needs to be published. It is an important paper to the fields of remote chemistry, and aerosol formation marine regions.*

*The reporting is descriptive, succinct, and easy to follow. The analytical and measurement methodologies used are all robust and generally have been previously wellproven. The analyses are performed to an appropriate level of detail, the conclusions drawn are well-supported, and the literature is comprehensively cited.*

*Therefore, I recommend publication with minor revision, and have only minor comments below.*

**We thank referee #2 for the helpful suggestions. We will address the comments in the following.**

*Specific comments:*

*Page 6, line 15 – If any species identifications beyond PFT identification by pigment (i.e. Figure 5) it would be helpful to point out whether they were species previously tested for isoprene production and present in Table 2 or not. This is particularly important in the areas where isoprene was not reproduced well.*

- **Unfortunately information on the species composition of diatoms has only been analyzed at ANT-XXV/1 cruise (4 stations, surface samples) and SPACES/OASIS cruise (12 stations, 3 depths) in order to verify the calculations of PFT from HPLC pigment data (see Taylor et al., 2011 and Bracher et al. 2016, respectively). No information on species composition is known for the other PFT groups at those cruises and not at all for the ASTRA-OMZ cruise. *Skeletonema sp.*, *Nizschia sp.* and *Thalasiosira sp.* species of diatoms were found at the analysed stations of ANT-XXV/1 and *Thalassiosira sp.* and *Chaetoceros sp.* were observed during SPACES/OASIS. All of these species are known to produce isoprene (see Table 2). Using the mean production rates for the diatom species we measured along the cruise tracks we get an average value of 2.16 µmol (g chl-*a*)$^{-1}$ day$^{-1}$ for ANT-XXV/1 which is in a good agreement with the mean value of 2.54 µmol (g chl-*a*)$^{-1}$ day$^{-1}$ used in this model. Using the values for *Thalassiosira sp.* and *Chaetoceros sp.* measured during SPACES/OASIS gives a mean production rate of 4.86 µmol (g chl-*a*)$^{-1}$ day$^{-1}$, which is twice as much as used in the model. In principle this would lower the discrepancy between the model output and the measurements in general. However compared to other PFTs, during SPACES/OASIS the contribution by diatoms was very low (on average 7%,), except for two stations (around 26°S and 46°E): here diatoms contributed 67% and 34%, respectively, to the total phytoplankton biomass which were also the stations with the highest total chl-*a* conc. (~1.1 mg chl-*a* m$^3$), while the rest of the campaign was mostly between 0.1 to 0.5 mg chl-*a* m$^3$.**

*Page 6, line 20 – This should be Figure 3, not 2*

- **Done, changed to Figure 3**

*Page 7, line 40 – I agree the physiological conditions can be a major driver of emission rates. A review of the laboratory studies that investigated this issue show a large range of emissions. This subject is worth a brief review of the relevant literature (~2 sentences).*

- **The text is changed to: "This highlights the need to measure … under different physiological conditions. Emissions in laboratory culture experiments can vary depending on the growth stage of the phytoplankton species (Milne et al., 1995). Shaw et al. (2003) showed that the health conditions of the phytoplankton species directly influence the emission rates of isoprene when using phage-infected cultures. But also environmental stress factors, such as temperature and light, influence the ability of different species to produce isoprene (Shaw et al., 2003;Exton et al., 2013;Meskhidze et al., 2015)."**

*It is important to provide the caveat that the in-situ data provided is focused on three cruises in two regions of the oceans. It is a good test, but there are many regions that have not yet been tested with the updated model.*

- **This is absolutely right. We clarify this statement page 8, line 38: "Even though the improved model is tested in three widely different ocean basins, there are still different regions where the model should be tested with direct isoprene measurements to verify the model output."**

*There is mention of the time resolution of the assessment being insufficient to capture the phytoplankton and isoprene heterogeneity that can result in large blooms of isoprene-producing species, and thus contributing to the underestimate of air concentrations. A sensitivity study based on bi-weekly or weekly satellite assessments of chlorophyll, as compared to monthly, would be an interesting addition to the manuscript. While it may not be possible to obtain MLD data on these time scales, perhaps there are pigment dat. Reasonable assumptions could be made in a simplistic manner to check what the maximum relative increase possible is in oceanic concentrations, flux, and ambient air concentrations. This would help determine if resolution is really the issue, or if untested high-producing species are the dominant cause of underestimate.*

- **In order to test our results in light of this comment, we used 8-day mean satellite data for chl-*a* and weekly satellite wind speed data for comparison with the model output for monthly mean satellite data. Weekly MLD data was not available and, as the agreement between monthly mean and in situ measured SST data was already good (c, Figure 3), we ran the model using monthly mean SST and MLD data. The model output is shown in Figure A (cyan). The comparison shows that in general the model outputs do not differ significantly except in the bloom region (10°-20°N). From this figure it is clear that monthly mean satellite data cannot resolve rapid changes, such as short phytoplankton blooms. However, using weekly mean satellite data will lower the data coverage in this study by 46%.**

[Figure]

**Figure A: Figure 2 from the manuscript including the model output using weekly mean satellite data (cyan). Comparison of observed (black) and modeled seawater isoprene concentrations for the ANT-XXV/1 cruise. Model calculations were carried out using the ISO$_{PS05}$ model configuration, with monthly mean satellite data (blue) for chl-*a*, wind speed, SST, and MLD (climatology) and *in situ* shipboard measurements (red).**

Plotting the model output using monthly mean satellite data versus weekly mean satellite data (Figure B) clearly shows that the precision of the monthly mean data is good enough in areas where there are no/few phytoplankton blooms (-30°-10°N and 30°-50°N, blue colors, close to 1:1 line). In contrast, during a phytoplankton bloom (10°-30°N, red) ), averaging over the month smears the signal, leading to an inaccurate representation of the chl-*a* distribution (more scatter around the 1:1 line).

[Figure]

**Figure B: Model output using monthly mean satellite data versus weekly mean satellite data. Color code indicates different latitudes (blue colors: non-bloom areas, red color: bloom area). Small figure is a zoom of the modeled concentrations of less than 10 pmol L$^{-1}$ for better resolution.**

These results show that the model is giving reasonable results either using monthly mean or weekly mean satellite data. It is the choice of the user to run the model either with monthly mean satellite data to get good spatial data coverage or to run the model with weekly mean satellite data to get better temporal resolution.

To account for the reviewers suggestion, we added following text to page 6, line 28: "8-day mean chl-*a* and weekly wind speed satellite data (not shown) are also available and could lower the discrepancies to the *in situ* data. For this study, 8-day values were not useful for this region and time, due to cloud coverage (loss of 46% of data points). A compromise between the two would be to average the 8-day values over a larger area grid to increase the amount of satellite derived data, but this would lower the resolution and therefore the accurate comparison with the cruise track."

**References**

Bracher A., Soppa M. A., Loza S., Dinter T., Wolanin A., Bricaud A., Brewin R., Rozanov V., (2016) SynSenPFT: Synergistic retrieval of phytoplankton functional types from space: from hyper- and multispectral measurements. Talk at Living Planet Symposium 2016, 12 May 2016, Prague, Czech Republic

Exton, D. A., Suggett, D. J., McGenity, T. J., and Steinke, M.: Chlorophyll-normalized isoprene production in laboratory cultures of marine microalgae and implications for global models, Limnology and Oceanography, 58, 1301-1311, 2013.

Meskhidze, N., Sabolis, A., Reed, R., and Kamykowski, D.: Quantifying environmental stress-induced emissions of algal isoprene and monoterpenes using laboratory measurements, Biogeosciences, 12, 637-651, 10.5194/bg-12-637-2015, 2015.

Milne, P. J., Riemer, D. D., Zika, R. G., and Brand, L. E.: Measurement of Vertical-Distribution of Isoprene in Surface Seawater, Its Chemical Fate, and Its Emission from Several Phytoplankton Monocultures, Marine Chemistry, 48, 237-244, Doi 10.1016/0304-4203(94)00059-M, 1995.

Shaw, S. L., Chisholm, S. W., and Prinn, R. G.: Isoprene production by Prochlorococcus, a marine cyanobacterium, and other phytoplankton, Marine Chemistry, 80, 227-245, http://dx.doi.org/10.1016/S0304-4203(02)00101-9, 2003.

Taylor, B. B., Torrecilla, E., Bernhardt, A., Taylor, M. H., Peeken, I., Röttgers, R., Piera, J., and Bracher, A.: Bio-optical provinces in the eastern Atlantic Ocean and their biogeographical relevance, Biogeosciences, 8, 3609-3629, 10.5194/bg-8-3609-2011, 2011.

---

## Author Comment (AC2) · 26 Aug 2016

*General Comments: This article describes an evaluation of surface ocean isoprene predictions from a steady-state model using an extensive dataset of cruise data, remotely sensed satellite data, and box modeling. Although the topic of marine isoprene production isn't new, this work describes the most comprehensive evaluation to date using data from cruises spanning multiple years and oceanic regions. The methods are clearly described in the study, and figures effectively summarize the results. Beside some minor technical edits, the manuscript is very well written. My main critique of the article is the unevenness of the results; the oceanic concentrations are thoroughly evaluated while the discussion of the box modeling results are brief and overly suggestive. I'd suggest that the article be published after addressing the comments below.*

**We thank referee #1 for reviewing of this manuscript and for providing helpful comments. We will address the comments in the following.**

*Specific Comments:*

*1) After an comprehensive evaluation of the seawater isoprene concentrations from the various cruises which clearly indicates the model improvement from the inclusion of phytoplankton functional types and reduction in bacterial degradation, I found the box modeling section of the results lacking. The article describes the existence of measured isoprene concentrations in the atmosphere from at least two cruises, yet these measurements are simply averaged and put into three curves of a figure. From this simplified analysis, an important conclusion is drawn (there are missing oceanic sources of isoprene) that appears in the abstract and conclusion of the article. I'd suggest either this analysis be removed or preferably expanded to include an evaluation of the atmospheric isoprene concentrations along the ship tracks. Specifically, I think the study could be informed by a box modeling study that moves with the ship location in order to identify the temporal and spatial extent of any missing oceanic isoprene sources.*

- **We addressed the reviewer's suggestion to make the box model results more robust by directly comparing to air concentrations of isoprene over the ship's cruise track. Figure 8 is changed to the following:**

[Figure]

**Figure 8: 1-day mean measured (blue) and calculated (red) daytime isoprene mixing ratios (ppt) during SPACES/OASIS (2014) and ASTRA-OMZ (2015). Calculated isoprene air values were derived by using the sea-to-air flux, a marine boundary layer height of 800 m and the one hour atmospheric lifetime based on a simple box model approach for each individual measurement.**

- Paragraph 4 starting at page 9 line 31 is now changed as follows: "Using atmospheric isoprene concentrations measured in two of the three campaigns, we were able to use a top-down approach to calculate isoprene emissions in order to compare with the bottom-up flux estimates. We used a box model with an assumed marine boundary layer height of 800 m, which reflected the local conditions during the two campaigns. The only source of isoprene for the air was assumed to be the sea-to-air flux (emission) and the atmospheric lifetime was assumed to be determined by reaction with OH (chemical loss, 1 h). The sea-to-air flux was calculated by multiplying $k_{AS}$ with the measured isoprene concentration ($C_W$) in the ocean (eq. (3)). We assumed $C_A$ to be zero in order to have the highest possible sea-to-air-flux, following a conservative approach. The concentration outside the box was assumed to be the same as inside to neglect advection in to and out of the box. The resulting calculated steady-state isoprene air concentration for every box (1-day mean value of all individual measurements at daytime) is shown in Figure 8 (for a one hour lifetime it takes approximately 10 hours to achieve steady state). For comparison, the mean measured concentration of isoprene in the atmosphere during the two cruises is 2.5±1.5 ppt and therefore 45 times higher than the calculated isoprene air values. The measured concentrations match previously measured remote open ocean atmospheric

values (Shaw et al., 2003). We only used atmospheric measurements which were obtained during daytime (to reflect reaction with OH) and were not influenced by terrestrial sources. This was determined by omitting data points with concomitant high levels of anthropogenic hydrocarbons (concentrations of butane higher 20 ppt). Reported mean atmospheric lifetime estimates of isoprene range from minutes up to four hours, depending mainly on the atmospheric concentration of OH (Pfister et al., 2008). We calculate that for an estimated lifetime of 1 h and 4 h, a sea-to-air flux of at least 2000 nmol m$^{-2}$ day$^{-1}$ and 500 nmol m$^{-2}$ day$^{-1}$, respectively, is needed to reach the atmospheric concentration measured during SPACES/OASIS and ASTRA-OMZ, which is approximately 10-20 times higher than computed (even when assuming $C_A$ as zero). Recent studies showed that the measured fluxes of isoprene range from 4.6-148 nmol m$^{-2}$ day$^{-1}$ in June/July 2010 in the Arctic (Tran et al., 2013) to 181.0-313.1 nmol m$^{-2}$ day$^{-1}$ in the productive Southern Ocean during austral summer 2010/2011 (Kameyama et al., 2014). Despite these high literature values, it appears that the calculated fluxes cannot explain the measured atmospheric concentrations even when a conservative lifetime of 4 h is assumed."

Please note, we changed the wording of the abstract (page 1, line 22) to the following in order to more comprehensively address the problem: "These findings suggest that there is at least one missing oceanic source of isoprene and, possibly, other unknown factors in the ocean or atmosphere influencing the atmospheric values. The discrepancy between calculated fluxes and atmospheric observations must be reconciled in order to fully understand the importance of marine derived isoprene as a precursor to remote marine boundary layer particle formation."

*2) The study clearly shows that phytoplankton function types can affect seawater isoprene concentrations, yet a comparison of measured and satellite-derived phytoplankton function type is not well described. I'd suggest describing in more detail the meaning of "discrepancy less than 25%" (Page 8, Line 35) in terms of the different phytoplankton functional types and oceanic regions and how any of these discrepancies may affect the uncertainty in the global marine isoprene emission estimate.*

- The following figure S1, which we will provide in the supplement, shows the comparison between measured and calculated phytoplankton functional types.
  The text in the supplement will be as follows: "Figure S1 shows the comparison between the measured isoprene production rate and the isoprene production rate derived from the phytoplankton functional type (PFT)-parameterization by Hirata et al. (2011). The comparison shows very good linear correlation in less productive regions (dashed regression line) whereas it is not linear over the whole range of isoprene production rates. The parameterization is dependent on the chl-*a* concentration and figure S1 shows, fairly clearly, that the parameterization overestimates the PFT concentration and, therefore, the isoprene production rate (dotted regression line) in productive regions. The phytoplankton pigment data used in the parameterization of Hirata et al. (2011) is well distributed in the Atlantic Ocean, sparsely distributed in the Indian Ocean region of SPACES/OASIS, and there has been no data used for the parameterization in the region off to Peru where ASTRA-OMZ took place. This may also cause some discrepancies between the measured and

calculated values. But as these overestimated PFT values only account for 5% of our data set the overall coefficient of determination between the derived data using Hirata et al. (2011) and the measured isoprene production rate is 0.89."

Page 8, line 35 in the manuscript is changed to: "The quality of…(coefficient of determination: $R^2$=0.89, Figure S1, supplement) …."

[Figure]

**Figure S 1: Measured isoprene production rates versus parameterized isoprene production rates from three different cruises (black: ANT-XXV/1; blue: SPACES/OASIS; red: ASTRA-OMZ). The dashed line and dotted line represent the regression line of isoprene production rates between 0 and 10 pmol $L^{-1}$ $day^{-1}$ and higher than 10 pmol $L^{-1}$ $day^{-1}$, respectively. The solid line represents the 1:1 line.**

*Minor Comments:*

*1) Page 1, Line 39: the yr-1 needs a superscript*

- **Done.**

*2) Page 4, Line 19: should be "Table 2 of Taylor..."*

- **Done.**

**References**

Hirata, T., Hardman-Mountford, N. J., Brewin, R. J. W., Aiken, J., Barlow, R., Suzuki, K., Isada, T., Howell, E., Hashioka, T., Noguchi-Aita, M., and Yamanaka, Y.: Synoptic relationships between surface Chlorophyll-a and diagnostic pigments specific to phytoplankton functional types, Biogeosciences, 8, 311-327, 10.5194/bg-8-311-2011, 2011.

Kameyama, S., Yoshida, S., Tanimoto, H., Inomata, S., Suzuki, K., and Yoshikawa-Inoue, H.: High-resolution observations of dissolved isoprene in surface seawater in the Southern Ocean during austral summer 2010-2011, Journal of Oceanography, 70, 225-239, 10.1007/s10872-014-0226-8, 2014.

Pfister, G. G., Emmons, L. K., Hess, P. G., Lamarque, J. F., Orlando, J. J., Walters, S., Guenther, A., Palmer, P. I., and Lawrence, P. J.: Contribution of isoprene to chemical budgets: A model tracer study with the NCAR CTM MOZART-4, Journal of Geophysical Research: Atmospheres, 113, n/a-n/a, 10.1029/2007JD008948, 2008.

Shaw, S. L., Chisholm, S. W., and Prinn, R. G.: Isoprene production by Prochlorococcus, a marine cyanobacterium, and other phytoplankton, Marine Chemistry, 80, 227-245, http://dx.doi.org/10.1016/S0304-4203(02)00101-9, 2003.

Tran, S., Bonsang, B., Gros, V., Peeken, I., Sarda-Esteve, R., Bernhardt, A., and Belviso, S.: A survey of carbon monoxide and non-methane hydrocarbons in the Arctic Ocean during summer 2010, Biogeosciences, 10, 1909-1935, 10.5194/bg-10-1909-2013, 2013.

---

## Author Response (AR2)

**Editor**

*The paper under discussion received good and excellent ratings by the reviewers and it has been further improved by the applied revisions. Therefore, I am pleased to accept the revised manuscript for publication in ACP. A few minor technical corrections may help to further clarify the description of the steady-state model used to estimate the atmospheric isoprene concentration.*

*Minor technical corrections:*

1) *In the description of the box model, you make the statement '..for a one hour lifetime it takes approximately 10 hours to achieve steady state'. If the boundary conditions (emission rate, boundary-layer height, isoprene lifetime) are kept constant, I would expect that the steady state is reached in 2-3 hours. Please check.*

   **We checked the model and it takes 9.21 hours until the loss term reaches 99.99% of the constant inflow (sea-to-air-flux). Therefore we decided to use the concentration after 10 hours for the steady-state-concentration.**

2) *If I understand the description correctly, the atmospheric steady-state isoprene concentration Ca is simply given by the equation: Ca = (kAS*Cw)*tau/h, where tau is the atmospheric lifetime of isoprene and h is the marine boundary-layer height. It would be helpful for the reader to include the equation, which you have used, into Paragraph 4.*

   **Yes, you are right. We implemented the equation in the text (page 10, line 11) as follows:**

[revised manuscript text omitted]